# Defining Optimal Conditions for Tumor Extracellular Vesicle DNA Extraction for Mutation Profiling

**DOI:** 10.3390/cancers14133258

**Published:** 2022-07-02

**Authors:** Julia Elzanowska, Laura Berrocal, Beatriz García-Peláez, Marta Vives-Usano, Beatriz Passos Sebo, Joana Maia, Silvia Batista, Jaakko Teppo, Markku Varjosalo, Maria Carolina Strano Moraes, Miguel Ángel Molina-Vila, Bruno Costa-Silva

**Affiliations:** 1Champalimaud Physiology and Cancer Programme, Champalimaud Foundation, 1400-038 Lisbon, Portugal; julia.elzanowska@research.fchampalimaud.org (J.E.); beatriz.sebo@research.fchampalimaud.org (B.P.S.); joanatcmaia@gmail.com (J.M.); saabatista@gmail.com (S.B.); carolina.stranomoraes@research.fchampalimaud.org (M.C.S.M.); 2Laboratorio de Oncología/Pangaea Oncology, Hospital Universitario Quirón-Dexeus, 08028 Barcelona, Spain; lberrocal@panoncology.com (L.B.); bgarcia@panoncology.com (B.G.-P.); mvives@panoncology.com (M.V.-U.); 3Drug Research Program, Faculty of Pharmacy, University of Helsinki, 00014 Helsinki, Finland; jaakko.teppo@helsinki.fi; 4Molecular Systems Biology Research Group and Proteomics Unit, Institute of Biotechnology, University of Helsinki, 00014 Helsinki, Finland; markku.varjosalo@helsinki.fi

**Keywords:** extracellular vesicles (EVs), DNA, mutation profiling, liquid biopsy

## Abstract

**Simple Summary:**

We here performed a comprehensive comparison of commonly used approaches for EV-DNA extraction, by assessing DNA quantity, its quality, and its suitability for downstream analyses. We found that the tested methods resulted in different DNA yields and fragment sizes. As a consequence, we found that different EV-DNA extraction methods had an impact on the detection of specific mutations by qPCR and on the suitability of next-generation sequencing (NGS), which could impact downstream clinical applications.

**Abstract:**

(1) Background: Extracellular vesicles (EVs) have emerged as crucial players in the communication between cells in both physiological and pathological scenarios. The functions of EVs are strongly determined by their molecular content, which includes all bioactive molecules, such as proteins, lipids, RNA, and, as more recently described, double-stranded DNA. It has been shown that in oncological settings DNA associated with EVs (EV-DNA) is representative of the genome of parental cells and that it reflects the mutational status of the tumor, gaining much attention as a promising source of biomarker mutant DNA. However, one of the challenges in studies of EV-DNA is the lack of standardization of protocols for the DNA extraction from EVs, as well as ways to assess quality control, which hinders its future implementation in clinics. (2) Methods: We performed a comprehensive comparison of commonly used approaches for EV-DNA extraction by assessing DNA quantity, quality, and suitability for downstream analyses. (3) Results: We here established strategic points to consider for EV-DNA preparation for mutational analyses, including qPCR and NGS. (4) Conclusions: We put in place a workflow that can be applied for the detection of clinically relevant mutations in the EV-DNA of cancer patients.

## 1. Introduction

Extracellular vesicles (EVs) are lipid-bilayer-enclosed particles released in the extracellular milieu by virtually all cells [1]. In recent years, EVs have emerged as crucial mediators of cell–cell communication, acting as messengers in both physiological and pathological conditions [2,3]. The functional roles of EVs are strictly related to their molecular composition, which has been shown to be highly heterogeneous and to consist of a variety of biomolecules, including proteins, lipids, metabolites, and nucleic acids [3,4]. The discovery of DNA associated with EVs (EV-DNA) sparked an interest in its biological and clinical functions, and in recent years, multiple roles of EV-DNA have been explored in terms of intercellular communication, immune response modulation, and maintenance of cellular homeostasis [5,6,7,8,9,10]. Moreover, the ubiquitous presence of EVs in different body fluids makes them strong candidates for nucleic-acid biomarkers in liquid biopsies. Indeed, EVs have lately attracted attention as a potential source of biomarker DNA in various pathophysiological scenarios, especially in oncological contexts [11,12]. Several studies have shown promising results, supporting the idea that EV-DNA represents the entire genome of parental cells and that its analysis allows specific gene mutations, including clinically relevant alterations, to be detected [11,12,13,14]. These studies have indicated that EV-DNA is suitable for liquid biopsies, both for early diagnosis and for disease prognosis.

To date, various studies have addressed the standardization of RNA extraction from EVs (EV-RNA) and have provided key steps to be considered for the EV-RNA preparation for downstream analyses such as RNA-Seq [15,16]. These studies have shown that different RNA purification methods can indeed result in differences in RNA yield and quality. However, comprehensive comparisons of EV-DNA preparation for downstream approaches are still missing. In fact, the implementation of EV-DNA as a clinical biomarker is still halted by the lack of comprehensive studies on the technical aspects of different strategies for EV-DNA isolation, characterization, and quality control. The literature describes multiple EV-DNA extraction approaches that have been applied (Table 1). As DNA mutational analyses can be directly affected by DNA quantity and quality, in order to assure reproducibility as well as high confidence in the results, it is crucial that a standardized workflow for EV-DNA extraction and quality control is developed. Yet, thus far, multiple non-standardized EV-DNA extraction approaches have been employed (Table 1).

In this study, we established strategic points for EV-DNA preparation for mutational analyses. This included DNA extraction from EVs, its quantification and quality analysis, and the assessment of its suitability for downstream applications, including qPCR and next-generation sequencing (NGS). By using four different commercial kits, as well as a phenol–chloroform-based protocol, DNA was extracted from EVs isolated from the cell-conditioned media (CCMs) of a panel of cancer cell lines. Extracted EV-DNA was subjected to a comprehensive comparison of the methods, with a specific focus on their suitability for biomarker DNA analyses. Moreover, we present evidence about the utility of the established protocol to detect clinically relevant mutations in plasma samples of cancer patients. To our knowledge, this is the first study to compare the impact of the EV-DNA isolation protocol on sample quantity, quality, and suitability for mutational analyses.

## 2. Materials and Methods

### 2.1. Preparation of Conditioned Cell Culture Media—Cell Lines and Cell Culture

All cell lines were obtained from the American Type Culture Collection. Human lung cancer cell lines H1975, H1650, and HCC827 and human pancreatic cancer cell line PANC10.05 were maintained in RPMI 1640 (Corning). Human breast cancer cell line MDA-MB-231 (purchased from Memorial Sloan Kettering Cancer Center) and human colorectal cancer cell line SW480 were grown in DMEM (Corning). LoVo, a human colorectal cancer cell line, was cultured in F12-K (Corning), and human pancreatic cancer cell line CFPAC-1 was cultured in IMDM (Gibco). All the above-mentioned media were supplemented with 1% penicillin–streptomycin (Gibco 15-140-122; Waltham, MA, United States) and 10% fetal bovine serum (FBS) (Biowest S181BH-500; Nuaillé, France), except for PANC10.05, which was grown in medium supplemented with 15% FBS (Biowest S181BH-500; Nuaillé, France) and 10 U/mL insulin (Gibco). Cells were maintained at 37 °C at 5% CO_2_ levels. For conditioning, cells were cultured in the designated media supplemented with 1% penicillin–streptomycin and 10% EV-depleted FBS [26]. For the preparation of conditioned culture media (CCMs), cells were seeded in 150 mm culture dishes containing 20 mL of EV-depleted medium and grown for 72 h until they reached a confluency of ~90%. The number of cells seeded for each cell line at the initiation of culture is presented in Appendix A. Non-conditioned medium containing 10% EV-depleted FBS was used as a control. After conditioning, CCMs and the non-conditioned medium were immediately submitted to two centrifugation steps (500× *g* for 10 min and 3000× *g* for 20 min, both at 10 °C) to remove dead cells and large debris; then, they were either directly used for EV isolation or stored at −80 °C until use. The cell lines used as positive and negative controls for qPCR are presented in Appendix A.

### 2.2. Plasma Samples and Ethics Statement

To obtain plasma, blood samples were collected from patients during daytime in sterile Vacutainer tubes (BD) containing EDTA. Blood was directly subjected to 2 consecutive centrifugation steps (500 g, 10 min) and visually inspected for obvious signs of hemolysis. Plasma samples were then aliquoted and frozen at −80 °C. The clinical characteristics of patients and technical aspects of sample collection are summarized in Appendix A, respectively.

The study was carried out in accordance with the principles of the Declaration of Helsinki under an approved protocol of Dexeus Hospital Ethical Committee (52/2018 from 22 November 2018). This research study was approved by Research Ethics Committee of Quiron Salud Hospital Group in Barcelona on 25/10/2018 (ethic code: 51/2018). Informed consent was obtained from all patients; samples were deidentified for patient confidentiality.

### 2.3. Extracellular Vesicle Isolation and Characterization

#### 2.3.1. EV Isolation by Ultracentrifugation

All EV samples were isolated by differential ultracentrifugation combined with a sucrose cushion, as previously described [27]. For each analyzed cancer cell line, at least 3 independent batches of EVs from separately grown cultures were isolated. First, the CCM was centrifuged at 12,000× *g* for 20 min to remove larger EVs, including apoptotic bodies. The supernatant (CCM and plasma) was then centrifuged at 100,000× *g* for 2 h 20 min. The pellet was resuspended in 16 mL of filtered phosphate-buffered saline (PBS; Corning 15313581; New York, NY, USA) and added to the top of a 4 mL sucrose solution (D2O containing 1.2 g of protease-free sucrose and 96 mg of Tris base adjusted to pH 7.4). Next, centrifugation at 100,000× *g* for 1 h 10 min was performed; then, a 4 mL fraction was collected using a 18 G needle and mixed with 16 mL of filtered PBS. After overnight centrifugation at 100,000× *g*, the pellet of purified EVs was resuspended in filtered PBS. All used solutions were filtered using 0.22 μm filters. All centrifugation steps were performed at 10 °C using 45Ti or 70Ti rotors (Beckman-Coulter, Brea, CA, USA).

#### 2.3.2. Nanoparticle Tracking Analysis (NTA)

To determine particle concentration and size distribution, all EV samples were analyzed by NTA using NanoSight NS300 (Malvern Instruments, Malvern, UK) with a red laser (638 nm). Samples were diluted with filtered PBS to obtain concentrations within the optimal range for NTA analyses. The videos were recorded using a camera level of 16 and a threshold of 5-7 and further analyzed with NTA software v3.4 (Malvern Instruments, Malvern, UK).

#### 2.3.3. Protein Quantification

Protein concentration in EV samples was assessed using BCA Protein Assay Kit (Thermo Fisher Scientific, Waltham, MA, USA) according to the manufacturer’s instructions.

#### 2.3.4. Western Blotting

Western blotting was used to assess the presence of EV and non-EV protein markers. Equal protein amounts of EV samples were mixed with 4X Laemmli buffer (Bio-Rad), denatured for 5 min at 95 °C, and loaded onto 4–20% Mini-PROTEAN TGX Stain-Free Protein Gels (Bio-Rad). SDS-PAGE was run for 1.5 h at 90 V; then, proteins were transferred to nitrocellulose membranes (Cytiva) at 100 V for 1 h. Membranes were blocked with LI-COR Intercept Blocking Buffer (LI-COR Biosciences, Bad Homburg, Germany) for 1 h at RT. Blocked membranes were incubated overnight at 4 °C with primary antibodies diluted in LI-COR blocking buffer with 0.1% Tween-20. Membranes were washed with TBS-T (TBS with 0.1% Tween-20) three times for 5 min and then incubated with secondary antibodies for 1 h at RT. Incubation was followed by three additional washes with TBS-T of 5 min each. Blots were imaged using the Odyssey Infrared Imaging System (LI-COR Biosciences). The detailed list of primary and secondary antibodies used is provided in Appendix A. Original images shown in Appendix A. 

#### 2.3.5. Transmission Electron Microscopy

The samples were analyzed by negative staining. In brief, the samples were fixed with 2% Formaldehyde in Phosphate Buffer for 5 min. Fixed samples were incubated for 5 min on glow-discharged, formvar-coated, carbon-coated 200-mesh copper grids prior to contrasting with 2% aqueous Uranyl Acetate for 5 min. Imaging was performed on a Tecnai G2 Spirit BioTWIN transmission electron microscope equipped with an Olympus-SIS Veleta CCD camera.

### 2.4. Mass Spectrometry

Liquid chromatography–mass spectrometry (LC-MS) was used to assess the presence of EV and non-EV protein markers. To extract and solubilize the proteins, 9 mol/L urea (Sigma-Aldrich) was added to the final concentration of 8 mol/L, and the samples were sonicated for 15 min. Insoluble debris was removed by two 15 min rounds of centrifugation at 21,460× *g* at RT. To dilute the samples’ urea concentration to 1.5 mol/L, 50 mmol/L ammonium bicarbonate (Fluka) was added. Proteins were reduced and carbamidomethylated with dithiothreitol (Sigma-Aldrich; 5 mmol/L final concentration, 60 min at 37 °C) and iodoacetamide (Sigma-Aldrich; 15 mmol/L final concentration, 30 min at RT in the dark), respectively, followed by overnight digestion using 1 µg of sequencing-grade modified trypsin (Promega). For C18 purification, 10% trifluoroacetic acid (VWR) was added to the samples to a final concentration of 0.5% and acetonitrile (Merck) to the final concentration of 1%. C18 purification was performed with MicroSpin columns (The Nest Group, Inc.) as previously described [28]. Finally, the samples were vacuum-centrifuged to dryness and stored at −80 °C until LC-MS analyses.

LC-MS analyses were carried out with EASY-nLC 1000 coupled to an Orbitrap Elite mass spectrometer (both from Thermo) using LC-MS-grade solvents (water (VWR); acetonitrile (Merck)) and acids (VWR). The peptides were resolubilized in eluent A (1% acetonitrile + 0.1% formic acid + 0.01% trifluoroacetic acid in water), and 4 µg of total digested protein was injected. Thermo Acclaim PepMap C18 columns were used for separation at a flow rate of 300 nL/min (trap column: PepMap 100, 75 m × 2 cm, 3 m, 100; analytical column: PepMap RSLC, 75 m × 15 cm, 2 m, 100 Å). The run time was 140 min, 120 min of which were spent gradienting from 5% to 35% eluent B (98% acetonitrile + 0.1% formic acid + 0.01% trifluoroacetic acid in water). Nanospray voltage was 2.9 kV. MS detection was performed with Top20 data-dependent acquisition, where the 20 most intense ions from each MS1 full scan were isolated, fragmented, and analyzed in the ion trap. Then, 30 s dynamic exclusion was applied. The LC-MS parameters have been previously described in detail [28]. MaxQuant/Andromeda version 1.6.1.0 with standard settings was used for protein identification and label-free quantification. The database was a UniProt/Swiss-Prot reviewed human proteome. A false discovery rate of 1% was used on peptide and protein levels. Decoy hits, potential contaminants, and proteins identified only by modified peptides were removed, and LFQ intensities were used for estimating protein abundance.

### 2.5. DNA Extraction

Total DNA associated with EVs was extracted from previously isolated EVs (differential ultracentrifugation) using different DNA isolation methods: QIAamp DNA Mini Kit (Qiagen), QIAamp DNA Micro Kit (Qiagen), SeleCTEV Exosomal DNA Kit (Exosomics), XCF Exosomal DNA Isolation Kit (SBI), and phenol–chloroform extraction. DNeasy Blood & Tissue Kit (Qiagen), which is the most commonly applied approach for EV-DNA extraction in the literature (Table 1), was not included in this comparison as it is based on the same technology and used for the same purposes as the QIAamp DNA mini kit (Qiagen), which is our in-house DNA extraction protocol. In order to extract DNA using the SeleCTEV Exosomal DNA kit (Exosomics), we used only the part of the protocol dedicated to DNA purification. For DNA extraction using all methods, each independent EV batch was split, and equal protein amounts of EVs were used for each extraction method (Graphical Abstract). For commercial kits, extraction was performed as per the manufacturer’s instructions. For phenol–chloroform extraction, EVs were first mixed with 400 µL of lysis buffer (0.05 M EDTA, 0.01 M Tris-HCl (pH 8.0), 0.1 M NaCl, 10% SDS, and 200 µg/mL proteinase K) and incubated for 1 h at 56 °C. Secondly, deproteinization was performed by the addition of an equal volume of phenol–chloroform–isoamyl alcohol mixture (25:24:1), followed by vortexing and centrifugation at 16,000× *g* for 5 min at RT. Next, the upper aqueous phase was separated from the organic phase, mixed with an equal volume of chloroform, and then mixed and centrifuged again at 16,000× *g* for 5 min at RT. The upper phase containing DNA was carefully collected and mixed with one-tenth volume of 3 M sodium acetate (pH 5.2), glycogen (to a final concentration of 0.1 µg/mL), and 2.5 volumes of cold absolute ethanol. The mixture was incubated overnight at −20 °C and then centrifuged at 16,000× *g* for 30 min at 4 °C. Next, the supernatant was discarded, and the pellet was rinsed with 70% ethanol and centrifuged at 16,000× *g* for 15 min at 4 °C. Finally, the supernatant was removed, and the pellet was left to air dry for 5 min and then resuspended in 50 µL of Mili-Q water. For the extraction of genomic DNA from cell lines, either the QIAamp DNA mini kit (Qiagen) or the QIASymphony DSP Virus/Pathogen midi kit (Qiagen) was used, and the extraction was performed as per the manufacturer’s instructions. For the extraction of DNA from plasma-derived EVs, the SeleCTEV Exosomal DNA kit (Exosomics) was used according to the manufacturer’s instructions. All EV-DNA samples were stored at −20 °C.

### 2.6. DNA Quantification and Quality Analysis

#### 2.6.1. Qubit Fluorometer 3.0

The Qubit quantification of DNA was performed using a Qubit fluorometer (3.0; Invitrogen) and Qubit dsDNA HS Assay Kit (Invitrogen) according to the manufacturer’s instructions. A volume of 2 µL of each EV-DNA sample was diluted in 198 µL of Qubit working solution.

#### 2.6.2. Fragment Analyzer

The quantification and sizing of DNA was performed by capillary electrophoresis on a fragment analyzer (Agilent) using the High Sensitivity NGS Fragment kit (Agilent). The analyses were performed as per the instructions provided by the manufacturer.

### 2.7. DNA Mutational Analysis

#### 2.7.1. PNA-Q-PCR (TaqMan) for Mutation Testing

EV-DNA and gDNA samples were analyzed for the presence of EGFR (exons 19, 21 and p.T790M), KRAS (exons 12, 13) and BRAF (exon 15) mutations using TaqMan. The assay is based on quantitative real-time PCR (qPCR) in the presence of a PNA clamp (Eurogentec, Seraing, Belgium) designed to inhibit the amplification of the wild-type alleles [29,30,31]. The assay has been fully validated, has an ISO15189 accreditation, and allows the absolute and relative abundances of mutant alleles to be estimated in positive samples. Amplification was performed in a final volume of 12.5 μL, using 6.25 μL of Genotyping Master Mix (Applied Biosystems, Foster City, CA, USA), 0.96 pmol of each primer, 1.2 pmol of probes, and 6.25 pmol (for exons 21 and p.T790M) or 62.4 pmol (for exon 19) of PNA. Purified EV-DNA and gDNA were used in the following amounts: 3 μL (4.5 ng) for BRAF and exon 21 analysis, 1 μL (1.5 ng) for exon 19 and all exons in KRAS, and 1 μL (1 ng) for p.T790M analysis. Using QuantStudioTM 6 Real-Time PCR System (Applied Biosystems/Thermo Fisher Scientific), samples were subjected to 50 cycles of 15 s at 92 °C and 1.5 min at 60 °C. The sequences of the primers and probes used are listed in Appendix A. Analyses were carried out in duplicates using one sample of purified EV-DNA or gDNA. In addition, all samples were analyzed in the absence of PNA to confirm the presence of purified EV-DNA and gDNA (WT). Genomic DNAs from cell lines at 1.5 ng/μL were used as positive and negative controls (Appendix A). A sample was considered positive if the same mutant allele was amplified in the two duplicates in the presence of PNA. If amplification was only detected in one duplicate, samples were reanalyzed and considered positive if at least one of the duplicates was positive for the same mutated allele.

#### 2.7.2. NGS for Mutation Testing

EV-DNA from the H1975 cell line was analyzed by NGS performed with GeneRead^®^ QIAact Pangaea Solid Tumor Custom Panel on plasma (Qiagen), which targets 20 genes frequently altered in cancer (Appendix A). DNA (40 ng) purified from EV-DNA samples was used as a template; clonal amplification was performed on 625 pg of pooled libraries, and following bead enrichment, the GeneReader (Qiagen) instrument was used for sequencing. Qiagen Clinical Insight Analyze (QCI-A) software was employed to align read data and call sequence variants, which were imported into the Qiagen Clinical Insight Interpret (QCI-I) web interface for data interpretation and the generation of the final custom report. Gene copy numbers were determined by QCI-A and QCI-I software according to an “in-house” algorithm.

### 2.8. Statistical Analysis

One-way ANOVA was applied to calculate the *p*-value. *p*-values of <0.05 were considered statistically significant. Statistical analyses and calculations of coefficients of variation were performed using GraphPad Prism software (GraphPad software).

## 3. Results

### 3.1. Characterization of EVs Isolated from Conditioned Media

Isolated EVs were characterized by several approaches. First, NTA was used to determine the concentration and size distribution of nanoparticles in isolated EV samples. Representative histograms of the frequency of particle size in EV samples derived from two representative cell lines, PANC10.05 and CFPAC-1, are shown in Figure 1A,B, respectively. To confirm the expression of EV markers (tetraspanins CD9 and CD81, Alix) as well as the absence of non-EV markers (Calnexin and GM130), Western blotting was performed (Figure 1C). Transmission electron microscopy (TEM) revealed the presence of negative-stained EVs, seen as cup-shaped structures (Figure 1D). Additionally, the purity of our isolated EVs was supported by the LC-MS analyses of a panel of proteins differentially present in or absent from small EVs (Figure 1E). Although tetraspanins CD63 and CD81 are commonly enriched in EVs, CD63 was not identified in the LC-MS analyses of our EV samples. Similarly, CD81 was not detected in any of the analyzed EV samples, despite being identified in the Western blot analysis of CFPAC-1 EVs. CD63 and CD81 are membrane proteins and are both hydrophobic (difficult to solubilize) and low-abundant, which makes them difficult to detect with our LC-MS method. Moreover, the scan speed of any mass spectrometer is not sufficient to detect all peptides in a sample, so some peptides cannot be detected even if they are present in sufficient amounts. If the peptides from these proteins elute from LC and enter MS at the same time with many other peptides that are present in higher concentrations, these higher-abundant peptides are prioritized in MS identification, and the lower-abundant ones are missed. Interestingly, a comprehensive analysis of EV protein markers across more than 100 different human tumor and non-tumor cell lines showed that only 40% of them were positive for CD63 [32].

### 3.2. Preparation of EV-DNA for Mutational Analyses

#### 3.2.1. Comparison of EV-DNA Extraction Methods

Previous reports have shown that EVs can carry DNA both in their lumen and on their surface [5,10,12,23]. However, as there is no consensus on which compartment contains DNA of diagnostic interest, we here define DNA associated with EVs as EV-DNA, independently of its location.

For the extraction of EV-DNA, five different methods frequently used in the literature were utilized (QIAamp DNA Mini Kit, QIAamp DNA Micro Kit, SeleCTEV, XCF Exosomal DNA Isolation Kit, and phenol–chloroform). These methods do not discriminate between luminal DNA and surface DNA, and as aforementioned, we considered that both were extracted. As such, these methods can be used in future studies focusing on either fraction. As outlined in the Graphical Abstract, at least three independent batches of EVs were isolated from each cell line; then, each batch was split into equal amounts, which were used to extract DNA with the five above-referenced methods. The amount of extracted dsDNA was quantified by Qubit, and this assessment showed that the use of different extraction methods resulted in different concentrations of DNA. Specifically, when normalized to samples purified with phenol–chloroform (the least efficient extraction method tested by us), EV-DNA samples extracted using the SeleCTEV Exosomal DNA kit had a significantly higher DNA yield than other methods (Figure 2). This includes the concentration of DNA in ng per μL (Appendix A) and in ng per particle (Appendix A) in samples with similar EV protein concentrations. Controls involving EV-DNA extraction (SeleCTEV) from corresponding volumes of non-conditioned media revealed that less than 7% of DNA isolated from CCMs may derive from components of the media (Appendix A). These results show that among the tested strategies, the SeleCTEV Exosomal DNA kit was the method that provided the highest DNA abundance. DNA concentration variability was also assessed for EV-DNA samples extracted using the different methodologies (Appendix A). We found that when all isolation methods and cells were simultaneously evaluated, SeleCTEV displayed the lowest coefficient of variation amongst all cells, reinforcing the application of this method for the isolation of EV-DNA.

DNA fragment size is another parameter that plays a key role in the downstream analysis of DNA sequences [33]. Based on that, we compared the size distribution of DNA fragments in EV-DNA using a fragment analyzer (Agilent). A representative electrophoretic histogram of EV-DNA extracted from MDA-MB-231 CCM is presented in Figure 3A. A comparison of the methods showed that the average size of DNA fragments varied among the samples extracted with different DNA extraction protocols, with the highest average sizes having been found in the EV-DNA samples extracted with the SeleCTEV Exosomal DNA kit and phenol–chloroform. However, in the case of phenol–chloroform, only 10 out of 27 extracted EV-DNA samples were successfully analyzed using the fragment analyzer due to the very low DNA concentrations in these samples (Figure 3B).

#### 3.2.2. Assessment of KRAS and EGFR Mutations in EV-DNA and gDNA by TaqMan

To understand the effect of EV-DNA extraction on its suitability for mutational analyses, qPCR (TaqMan) studies were performed. EV-DNA was analyzed to evaluate the mutation status of two genes known to be mutated in the tested cell lines from which EV-DNA samples were isolated, KRAS (MDA-MB-231, LoVo, SW480, CFPAC-1, and PAN10.05) and EGFR (H1975, H1650, and HCC827). For each cell line, three independent EV-DNA samples, previously isolated and analyzed for the comparison of EV-DNA extraction methods (Graphical Abstract), were chosen based on DNA concentration and used for TaqMan (a total of 24 EV-DNA samples per each method). As shown in Table 2, the investigated mutations were detected in 21/24 (87.5%), 22/24 (91.7%), 23/24 (95.8%), 22/24 (91.7%), and 14/24 (58.3%) EV-DNA samples extracted with the QIAamp DNA mini kit, QIAamp DNA micro kit, SeleCTEV Exosomal DNA kit, XCF Exosomal DNA Isolation kit, and phenol–chloroform, respectively. This demonstrates that the EV-DNA extraction method affected the suitability of DNA for mutation analysis by qPCR and that the use of the SeleCTEV Exosomal DNA kit resulted in the highest percentage of mutation detection among all methods. Extraction with the QIAamp DNA micro kit and XCF Exosomal DNA Isolation kit also resulted in high mutation detection in more than 90% of the analyzed samples. On the other hand, the mutations were undetectable in over 40% of samples extracted with phenol–chloroform. Moreover, the quantity of mutated DNA in EV-DNA versus gDNA of the corresponding cell lines was comparable regardless of the applied extraction method (Figure 4).

#### 3.2.3. Mutation Detection in EV-DNA by NGS

To further assess EV-DNA suitability for mutational analyses, we performed NGS using H1975-derived EV-DNA. EV-DNA extracted with all five DNA purification methods was analyzed using a customized panel covering 20 genes. Out of the five analyzed samples, only the one extracted with phenol–chloroform had insufficient material for NGS, resulting in an unsuccessful analysis. In contrast, the mutations present in the gDNA of the cell line from which the EVs were derived were successfully detected in all the EV-DNA samples extracted with the four commercial kits used. The highly similar quality parameters and the almost identical mutant allelic fraction values, representing the percentage of sequence reads carrying a mutant allele, both suggest that those four DNA purification methods allowed us to extract DNA of sufficient quantity and quality for comparable mutation detection by NGS (Table 3 and Table 4).

### 3.3. Plasma EV-DNA Mutation Detection by TaqMan

As stated above, DNA extracted using the SeleCTEV Exosomal DNA kit had a significantly higher DNA yield from CCM-derived EVs than any of the other methods analyzed in this study. In order to assess the usefulness of this approach to detect clinically relevant mutations in plasma samples, EVs were isolated from samples collected from cancer patients. Plasma-derived EVs were then characterized using NTA and Western blotting (Figure 5). Due to the limited plasma volume available, TEM was not performed for these samples. Next, DNA was extracted from these EV samples using the SeleCTEV Exosomal DNA kit. All mutations found in tissue biopsies and circulating free DNA were also identified in EV-DNA (Table 5).

## 4. Discussion

Although recent discoveries on EV-DNA and its potential as a biomarker have created excitement, there are still important technical aspects that need to be considered before its future application in clinics. One of the challenges in EV-DNA studies is the lack of the standardization of DNA extraction and quality control, which are crucial factors to be considered for downstream DNA analyses and can vary depending on the chosen DNA extraction method.

Here, we contributed to this standardization by evaluating different methodologies for extracting DNA from EVs. As a first layer of standardization, we fixed the EV isolation method used, so that all DNA would be extracted from comparable isolated EV samples. The chosen method for the isolation of EVs was differential ultracentrifugation coupled with a sucrose cushion, which is one of the most commonly used EV isolation techniques and is considered a gold standard for EV isolation. Other EV isolation methods were not compared, as it was beyond the scope of this paper [34].

In the literature, the characterization of extracted EV-DNA is rather superficial, with DNA quantification being performed using different methods and often presented in various units, making it very difficult to reproduce and compare results. Some of the most commonly applied methods used to assess DNA yield include spectrophotometry-based measurements on NanoDrop spectrophotometers (ThermoScientific), capillary-electrophoresis-based DNA analysis on a fragment analyzer (Agilent) or 2100 Bioanalyzer (Agilent), and methods based on fluorescent dies, such as Qubit (Life Technologies) or QuantiFluor (Promega). These DNA quantification methods are characterized by different lower and upper limits of detection, as well as different detection size ranges; thus, they need to be considered carefully depending on the analyzed sample type. For instance, NanoDrop has low sensitivity, reaching 1 ng/µL, and does not distinguish between single- and double-stranded DNA, while capillary-electrophoresis-based DNA quantification has a very narrow size detection range [35]. Based on previous reports that demonstrate that EVs carry nucleic acids in the order of ng with a broad size range (100 bp→10 kbp), all EV-DNA samples purified in this study were quantified using the HS DNA kit on Qubit, which has a quantitative range of 10 pg/μL–100 ng/μL [11,12,18,21,36]. According to the results presented by the manufacturer, DNA kits for Qubit are highly selective for dsDNA over RNA and ssDNA, as quantification is based on the fluorescence intensity of the fluorescent dye binding to dsDNA and not on UV absorbance. Thus, we propose that future studies could benefit from using Qubit for EV-DNA quantification.

Most of the data described in this study were obtained from EVs isolated from CCMs, and the conditioning was always performed in media supplemented with EV-depleted FBS. The use of EV-depleted FBS enabled tumor cell growth to be obtained and EVs to be produced in CCMs while avoiding, amongst others, cellular stress, changes in phenotype, and potential alterations in EV cargo packaging that might be caused by the full absence of FBS. Of note, it was recently proposed that FBS supplementation may result in residues of bovine DNA in CCM preparations [37]. We here found that EV-DNA extraction from volumes of non-conditioned media that corresponded to those utilized for EV isolation from CCMs displayed residual amounts of DNA (less than 7% of the EV-DNA obtained from CCMs). Importantly, as all culture media were prepared with the same lot of EV-depleted FBS, residual bovine DNA would in principle be equally present in all CCM samples that contained the same amount of FBS. Although the culture medium of PANC10.05 contained more than 10% FBS (15%), we did not find any noticeable increment in DNA yield in EVs isolated from PANC10.05 when compared to other cells. Together, these results indicate that in our experimental settings, the traces of bovine DNA in EV-depleted FBS were not sufficient to significantly impact the amount of EV-DNA in CCMs.

When comparing EV-DNA isolation strategies, we observed that there were differences in DNA abundance amongst samples extracted with different methods, agreeable with the studies performed with other nucleic acids, including EV-RNA [15,16]. Our data showed that the highest DNA concentrations were obtained using the SeleCTEV Exosomal DNA kit and the lowest concentrations with phenol–chloroform, suggesting that the SeleCTEV Exosomal DNA kit may be the method of choice if the highest abundance of DNA is considered.

In terms of the size distribution of EV-DNA fragments, we showed that the average sizes of DNA fragments also differed among the analyzed DNA extraction methods, with the average being ~2000 bp. Both the SeleCTEV Exosomal DNA kit and phenol–chloroform extractions resulted in the highest average sizes of DNA fragments. However, despite the high average size of EV-DNA in the case of phenol–chloroform, the efficiency of extraction was lower, leading to lower quantities of DNA. Thus, only a few of the phenol–chloroform extracted samples amounted to a concentration high enough to be detected by the fragment analyzer for sizing evaluation.

The different quantities and sizes of EV-DNA observed in our studies likely resulted from the potential differences in the extraction methods used. Among the analyzed methods, apart from phenol–chloroform, the remaining four are based on spin-column technology, with two of the kits (SeleCTEV Exosomal DNA Kit and XCF Exosomal DNA Isolation Kit) being specifically dedicated to exosomal DNA extraction. However, the details distinguishing them from regular DNA extraction kits are not included in the descriptions provided by the manufacturers. Both the QIAamp DNA micro kit and the QIAamp DNA mini kit are based on silica-membrane DNA purification. The main difference between these two approaches is that the QIAamp DNA micro kit is dedicated to small-volume samples and includes the addition of carrier RNA, which increases DNA binding to the membrane of the spin column and subsequently enhances the recovery of DNA. Furthermore, we also observed differences in DNA amounts amongst cell lines (Appendix A), which is in line with existing reports suggesting a differential abundance of DNA associated with EVs in various cancer models [12].

Previous studies have shown that DNA in EVs not only covers the entire genome of parental cells, but also allows identical mutations from the cells they are derived from to be detected, including clinically relevant mutations such as KRAS mutations in pancreatic cancer or BRAF in melanoma [11,12,13,14,38]. Thus, in order to compare the suitability of the EV-DNA extracted with all five methods for mutational studies, we analyzed all EV-DNA samples for the presence of mutations known to be present in their parental cells. Mutation detection using TaqMan differed among EV-DNA samples extracted with different methods, with the highest detection rate being observed using the SeleCTEV Exosomal DNA kit (Exosomics) (95.8%). This result is consistent with the highest DNA concentration and the largest average DNA size of EV-DNA extracted with this method, both of which are factors that can directly affect the TaqMan reaction. In addition, consistently with previous results regarding EV-DNA quantification, respective mutations were detected in less than 60% of samples extracted with phenol–chloroform. qPCR-based amplifications, such as TaqMan, are highly influenced by the quantity, quality, and purity of DNA samples, and the low purity of EV-DNA samples extracted with phenol–chloroform could be a reason for unsuccessful amplification. Phenol–chloroform-based methods use hazardous organic solvents and may result in the presence of PCR inhibitors such as divalent cations or proteins, directly affecting TaqMan analysis. On the contrary, column-based DNA isolation methods are described to yield DNA samples of higher purity, which could explain the differences in mutation detection amongst the compared methods.

Further, we verified the applicability of our strategy for the detection of mutations in EVs derived from the plasma of cancer patients. As pre-analytical steps of plasma processing can affect EV recovery, we recorded a list of parameters describing sample collection and processing. Previous studies have suggested that the presence of lipoprotein and platelet fragments can interfere with molecular studies of plasma EVs [39,40,41]. Therefore, although our plasma EV samples were not controlled for the potential presence of lipoproteins and platelets fragments, we propose that future studies are necessary to address whether these parameters can impact the study of EV-DNA composition. In order to extract DNA from isolated plasma EVs, we used the SeleCTEV Exosomal DNA kit, which had a significant higher DNA yield in CCM-derived EV analysis. It was previously suggested that SeleCTEV’s proprietary peptide improves plasma EV isolation for DNA studies when compared to ultracentrifugation [20]. Therefore, future research should consider the potential benefits of using SeleCTEV’s proprietary peptide for improved plasma EV-DNA studies. Due to the limitation of the plasma samples’ volume availability, a side-by-side comparison of other DNA extraction methods was not possible in our study. Importantly, the analysis of EV-DNA from the plasma of cancer patients showed that our workflow allowed us to detect clinically relevant mutations in those samples, confirming previous reports on mutation detection in EVs [11,13,14,20].

Along with PCR-based analyses such as TaqMan, one of the most commonly used methods to study mutations in EV-DNA is sequencing. To assess the suitability of EV-DNA samples for sequencing, samples were analyzed with a custom NGS panel. Apart from phenol–chloroform, all the other tested samples were successfully analyzed by NGS, which is in accordance with our previous results demonstrating that the purity of EV-DNA extracted with phenol–chloroform can impede their suitability for mutation detection analyses. In contrast with TaqMan results, which showed that the choice of different DNA isolation methods resulted in differences in mutation detection, NGS analyses indicated that EV-DNA extracted with the four tested commercial kits had very similar results regarding mutation detection and quality parameters. These results show that while SeleCTEV contributes to the improved analysis of EV-DNA by qPCR, no clear advantages amongst the four tested commercial kits were observed in our NGS studies. Recent reports have suggested that this may be due to the high sensitivity of NGS [42].

## 5. Conclusions

Collectively, our results show that the quantity and quality of EV-DNA can vary depending on the applied method of extraction and highlight the importance of the standardization of EV-DNA preparation for improved reproducibility of results. Our work provides strategic points to consider when preparing EV-DNA samples for downstream applications, including qPCR and NGS (Table 6). In agreement with previous reports, we found that EV-DNA reflects the mutational status of parental cells. Therefore, our data support the use of EV-DNA for the detection of tumor mutations [11,12,13,17].

## Figures and Tables

**Figure 1 cancers-14-03258-f001:**
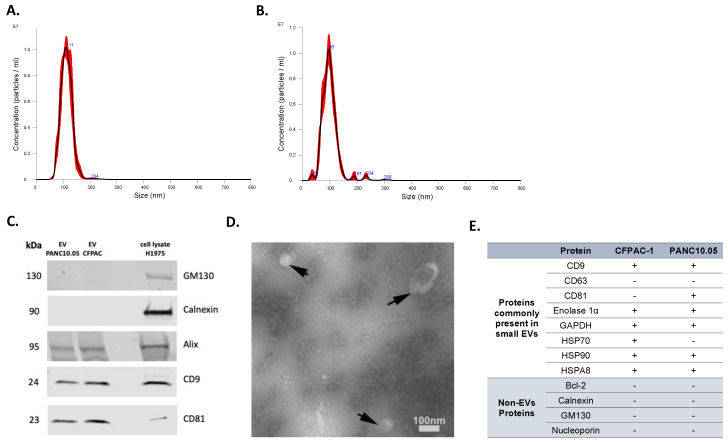
Characterization of EVs isolated from CCMs. Representative NTA histograms of EV samples isolated from (**A**) PANC10.05 and (**B**) CFPAC-1 CCM. (**C**) Representative Western blots of EV markers CD9, CD81, and Alix, and of non-EV markers Calnexin and GM130 for EV samples isolated from CCMs derived from CFPAC-1 and PANC10.05 cell lines (H1975 cell lysate was used as a control). (**D**) Representative TEM image of CCM-derived EV sample. (**E**) MS analysis of EV-positive and -negative markers of CFPAC-1 and PANC10.05-derived EVs.

**Figure 2 cancers-14-03258-f002:**
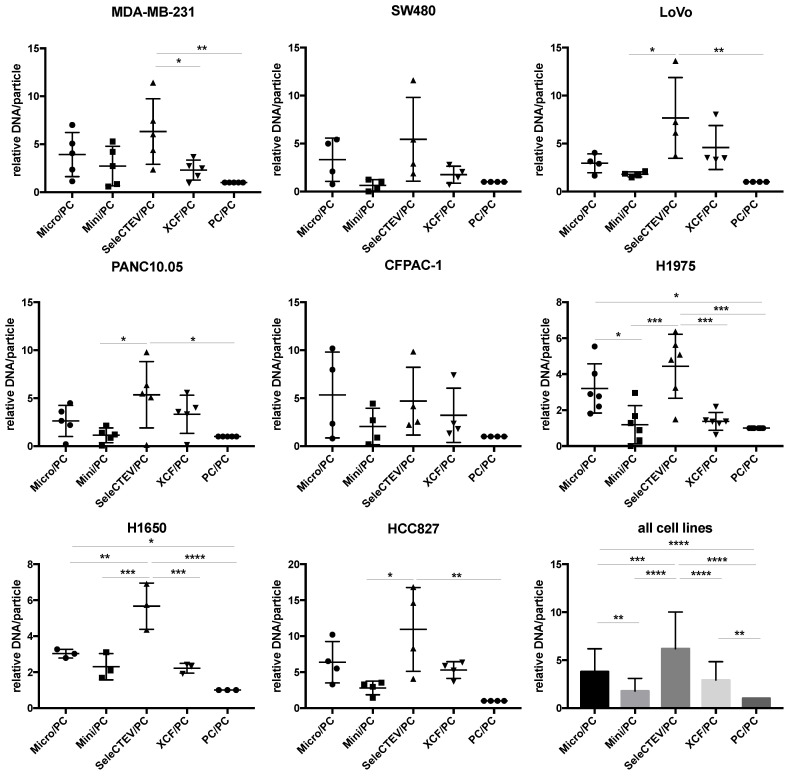
Comparison of EV-DNA abundance among different DNA extraction methods. Qubit DNA quantifications of EV-DNA samples extracted from a panel of cancer cell lines (MDA-MB-231, SW480, LoVo, PANC10.05, CFPAC-1, H1975, H1650, and HCC827) using QIAamp DNA Micro Kit (Micro), QIAamp DNA Mini Kit (Mini), SeleCTEV Exosomal DNA Kit (SeleCTEV), XCF Exosomal DNA Kit (XCF), and phenol–chloroform (PC). At least 3 independent batches of EVs were isolated from each cancer cell line. Next, each EV batch was split in equal amounts and used for DNA extraction with the indicated different methods. EV-DNA abundance is presented as ng of DNA/particle normalized to values obtained using phenol–chloroform. All the data are shown as the mean values ±SD from independent measurements. *p*-values were calculated using one-way ANOVA with Tukey’s post hoc test; **** *p* < 0.0001, *** *p* < 0.001, ** *p* < 0.01, * *p* < 0.05.

**Figure 3 cancers-14-03258-f003:**
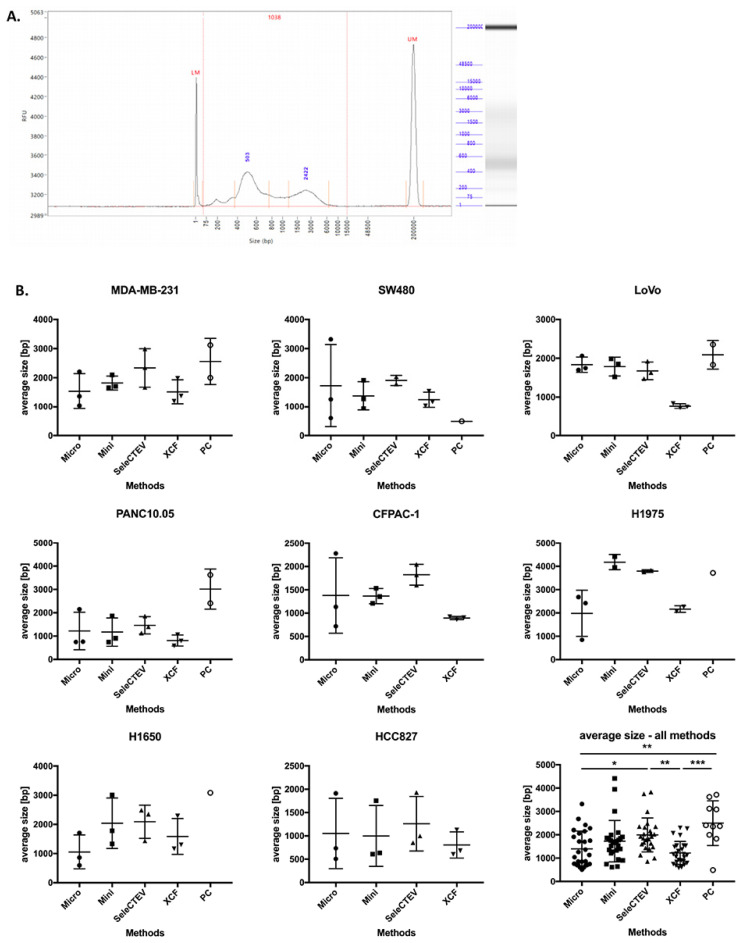
Comparison of the average size of EV-DNA fragments among different DNA extraction methods. (**A**) Fragment-analyzer analysis of EV-DNA extracted from MDA-MB-231 CCM, showing size distribution (bp) and relative fluorescence intensity (RFU). The peaks at 1 bp and 20,000 bp correspond to 2 internal size markers. (**B**) Average size (bp) of EV-DNA fragments extracted with the following kits: QIAamp DNA Micro Kit (Micro), QIAamp DNA Mini Kit (Mini), SeleCTEV Exosomal DNA Kit (SeleCTEV), XCF Exosomal DNA Kit (XCF), and phenol–chloroform (PC). The data are presented as mean values ±SD from three independent specimens. *p*-values were calculated using one-way ANOVA with Tukey’s post hoc test; *** *p* < 0.001, ** *p* < 0.01, * *p* < 0.05. Note: Some samples extracted using phenol–chloroform could not be successfully analyzed using the fragment analyzer due to very low DNA concentrations.

**Figure 4 cancers-14-03258-f004:**
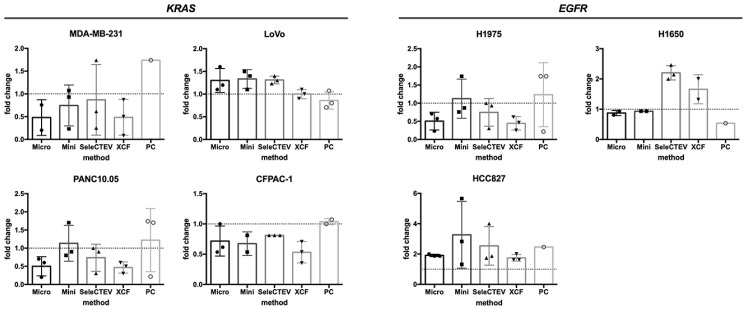
Mutational analyses of EGFR and KRAS genes in CCM-derived EV-DNA by TaqMan. Comparison of the quantity of detected mutated DNA in EV-DNA versus gDNA samples by TaqMan. The 2^−ΔΔCT^ method was used as a quantification strategy to assess the fold difference in mutated DNA in EV-DNA in relation to gDNA. This analysis could not be applied to SW480 EV-DNA as the WT KRAS gene was not amplifiable.

**Figure 5 cancers-14-03258-f005:**
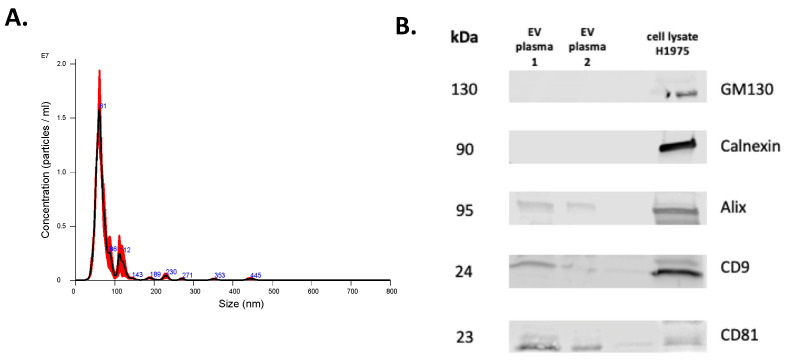
Characterization of EVs isolated from plasma. Representative NTA histogram of EV sample (**A**) and representative Western blots (**B**) of EV markers CD9, CD81, and Alix and of non-EV markers Calnexin and GM130 (H1975 cell lysate was used as a control).

**Table 1 cancers-14-03258-t001:** Summary of methods used for EV-DNA extraction in the literature.

Extraction Method	Manufacturer	Number of Times Used in the Literature	Reference
DNeasy Blood & Tissue Kit	Qiagen	3	[7,10,11]
QIAamp DNA Micro Kit	Qiagen	2	[14,17]
Phenol–Chloroform	N/A	2	[5,6]
QIAamp Circulating Nucleic Acid Kit	Qiagen	1	[18]
Qiagen Allprep DNA/RNA Mini Kit	Qiagen	1	[19]
QIAamp DNA Mini Kit	Qiagen	1	[12]
SeleCTEV Low Input DNA Enrichement Kit	Exosomics	1	[20]
TIANamp Genomic DNA Kit	Tiangen Biotech Co. Ltd.	1	[21]
Quick-gDNA Miniprep Kit	Zymo Research	1	[22]
Genomic DNA Mini Kit	Geneaid	1	[23]
XCF Exosomal DNA Isolation Kit	System Biosciences	1	[24]
Isopropyl alcohol precipitation	N/A	1	[25]
Proteinase K	Wako	1	[8]

**Table 2 cancers-14-03258-t002:** Detection of EGFR and KRAS mutations in cancer cells’ EV-DNA that were extracted using QIAamp DNA Micro Kit (Micro), QIAamp DNA Mini Kit (Mini), SeleCTEV Exosomal DNA Kit (SeleCTEV), XCF Exosomal DNA Kit (XCF), and phenol–chloroform (PC).

		EV-DNA Purification Method				
Gene	Cell-Line-Derived EVs	Micro	Mini	SeleCTEV	XCF	PC
*KRAS*	MDA-MB-231	2/3	3/3	3/3	3/3	1/3
	SW480	3/3	2/3	2/3	1/3	1/3
	LoVo	3/3	3/3	3/3	3/3	3/3
	PANC10.05	3/3	3/3	3/3	3/3	2/3
	CFPAC-1	3/3	2/3	3/3	3/3	2/3
*EGFR*	H1975	3/3	3/3	3/3	3/3	3/3
	H1650	2/3	3/3	3/3	3/3	1/3
	HCC827	2/3	3/3	3/3	3/3	1/3
Samples with detected mutations		21/24 (87.5%)	22/24 (91.7%)	23/24 (95.8%)	22/24 (91.7%)	14/24 (58.3%)

**Table 3 cancers-14-03258-t003:** Mutation detection in H1975 EV-DNA by NGS.

Method			Mini	Micro	SeleCTEV	XCF	PC
Gene	Exon	Mutation	Allelic fraction				
*EGFR*	21	c.2573T > G	66%	65%	67%	62%	N/A
	20	c.2369C > T	64%	64%	65%	69%	N/A
*TP53*	8	c.818G > A	98%	97%	97%	96%	N/A

**Table 4 cancers-14-03258-t004:** Summarized quality parameters of H1975 EV-DNA NGS analysis.

Method	Reads	Nucleotides	Average Read Length	Reads with Average Quality > 25	Reads Mapped	Reads in Target Regions	Base Positions in Regions of Interest with UMI Coverage > 100×	Base Positions in Regions of Interest with UMI Coverage > 60×
Mini	4023974	505992583	125.74	95.15%	3980643 (98.2%)	3557632 (89.37%)	97.68%	98.54%
Micro	4034733	512374900	126.99	94.89%	4023363 (99.72%)	3332780 (82.84%)	97.04%	97.55%
SeleCTEV	4194485	523511719	124.81	94.59%	4183430 (99.74%)	2398744 (57.34%)	96.75%	97.31%
XCF	4875693	127.70	94.93	94.93%	4868494 (99.85%)	4568724 (93.84%)	97.60%	98.28%
PC	N/A	N/A	N/A	N/A	N/A	N/A	N/A	N/A

**Table 5 cancers-14-03258-t005:** Mutational analysis of KRAS and BRAF genes in plasma-derived EV-DNA by TaqMan.

Patient	Tumor Type	Gene	Position	Mutant Allelic Fraction %
#1	Melanoma	BRAF	p.V600E	0.74 ± 0.04
#2	Colorectal	KRAS	p.G12V	20.33 ± 1.81
#3	Lung	KRAS	p.G12C	4.65 ± 0.83
#4	Lung	KRAS	p.G12D	4.84 ± 1.25
#5	Lung	KRAS	p.G12V	0.39 ± 0.55
#6	Lung	BRAF	p.V600E	0.84

**Table 6 cancers-14-03258-t006:** Strategic points for EV-DNA extraction and quality control for mutational analyses.

Extraction Method	DNA Yield	DNA Quality (Integrity)	Suitability for TaqMan	Suitability for NGS
QIAamp DNA Mini Kit	++	++	++	+++
QIAamp DNA Micro Kit	++	+++	++	+++
SeleCTEV Exosomal DNA Kit	+++	+++	+++	+++
XCF Exosomal DNA Isolation Kit	++	++	++	+++
Phenol–Chloroform	+	+++	+	-

+++—high; ++—medium; +—low; -—not suitable.

## Data Availability

The data presented in this study are available upon request from the corresponding author.

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
