# Peer review of "Defining Optimal Conditions for Tumor Extracellular Vesicle DNA Extraction for Mutation Profiling"

_cancers, 2022, doi:10.3390/cancers14133258_

Round 1

Reviewer 1 Report

Julia Elzanowska and colleagues presented this scientific work of high interest for the scientific community, especially researchers working on EVs as circulating biomarkers. This article looks an useful contribution to the standardization in EVs-methodologies, that is strongly required to  produce reproducible data on EVs- based liquid biopsy. However, some concerns on missing details : although authors have been detailed from EVs-DNA extraction on, many specifications are required on the condition of EVs recovery. In details, regarding cell lines-derived EVs, at least the number of cells used for the extractions should be declared. In patients characteristics, time of day of collection (Circadian variations) and patients therapy should be at least mentioned. Similarly, technical factors including fluid collection volume (how much blood? how much plasma?), first-tube discard, description of transport (if any), whether tube remained upright before processing, degree of hemolysis, possible confirmation of platelet and lipoprotein depletion prior to storage and other parameters should be clearly indicated (as raccomanded by MISEV guidelines 2018). If not possible, please add in the discussion a brief section on the importance of Pre-analytical steps in EVs yield, mentioning those parameters you could not collect as a limitation of the study.

In addition,  authors  should perform NTA, TEM and WB for EVs recovered from plasma samples.

Authors should add electron microscopy images mentioned in the title of the first result, but missing in the text and among figures.

Author Response

Reviewer 1:

We thank the reviewer for the constrictive suggestions. Point-by-point responses are provided bellow. Please also find attached the revised manuscript.

Julia Elzanowska and colleagues presented this scientific work of high interest for the scientific community, especially researchers working on EVs as circulating biomarkers. This article looks an useful contribution to the standardization in EVs-methodologies, that is strongly required to  produce reproducible data on EVs- based liquid biopsy. However, some concerns on missing details :

  1. although authors have been detailed from EVs-DNA extraction on, many specifications are required on the condition of EVs recovery. In details, regarding cell lines-derived EVs, at least the number of cells used for the extractions should be declared.

Reply: We added Supplementary Table 1 presenting the number of cells seeded per plate for conditioning of each cell line (2.1 Preparation of conditioned cell culture medium – cell lines and cell culture, Page 3, Line 113-114)

  1. In patients characteristics, time of day of collection (Circadian variations) and patients therapy should be at least mentioned. Similarly, technical factors including fluid collection volume (how much blood? how much plasma?), first-tube discard, description of transport (if any), whether tube remained upright before processing, degree of hemolysis, possible confirmation of platelet and lipoprotein depletion prior to storage and other parameters should be clearly indicated (as raccomanded by MISEV guidelines 2018). If not possible, please add in the discussion a brief section on the importance of Pre-analytical steps in EVs yield, mentioning those parameters you could not collect as a limitation of the study.

Reply: We added the missing information regarding patients' characteristics and technical aspects of sample processing in Supplementary Table 3 and Supplementary Figure 4 (2.2 Plasma samples and ethics statement,Page 3, Lines 122-127). Additionally, we added a section in Discussion (Page 17-18, Discussion, Lines 719-728) describing the importance of pre-analytical steps in EV analysis and mentioning one of the parameters we did not collect.

  1. In addition, authors  should perform NTA, TEM and WB for EVs recovered from plasma samples.

Reply: We added Figure 5 presenting representative NTA and WB of plasma-derived EVs (3.3 Plasma EV-DNA mutation detection by TaqMan, Page 15, Lines 597-608). As explained in the text, due to the limited availability of cancer patient derived plasma samples, TEM was not performed for these samples.

  1. Authors should add electron microscopy images mentioned in the title of the first result, but missing in the text and among figures.

Reply:

Representative TEM image of EV sample isolated from CCM was added to Figure 1 (3.1 Characterization of EVs isolated from conditioned media, Page 9-10, Lines 335-356; 416-445).

Reviewer 2 Report

In their manuscript titled “Defining optimal conditions to tumor Extracellular Vesicle DNA extraction for mutation profiling” Elzanowska et al. performed a comparative analysis of five different DNA extraction methods/kits for the extraction of EV-DNA from EVs that have been isolated by ultracentrifugation from conditioned cell culture media. The authors show that each method/kit has different advantages or disadvantages when it comes to DNA yield, DNA size, and suitability in downstream assays such as TaqMan qPCR and next-generation sequencing. The authors then use one method for DNA extraction from plasma EVs for use in mutation detection. This is a useful study for the EV field to help investigators choose the most appropriate DNA extraction methods for their studies; however, there are some outstanding questions that should be addressed.

Major concerns:

11.  One of the DNA extraction kits chosen by the authors, the SeleCTEV Exosomal DNA Kit, is designed not only to be used for DNA extraction from EVs but also for EV isolation from a biofluid. For the SeleCTEV kit the EV isolation is performed using an included proprietary peptide. It is unclear from the Methods section whether the authors included the peptide step in their DNA extractions or if they simply omitted the peptide-mediated EV isolation steps for their workflow. Moreover, it would be interesting (and important) to determine if the SeleCTEV Exosomal DNA Kit performs similarly for DNA isolation when the proprietary peptide that is included is used for EV isolation, rather than ultracentrifugation. This would be important information for the EV community as this kit could potentially be used for all steps – EV isolation and DNA extraction – in study workflows.

22. For their plasma samples and DNA mutation analysis the authors chose to only use the SeleCTEV kit for DNA extraction (Table 5). While their data show that DNA extraction with this kit permits identification of all mutations present in the tumour tissue, this analysis would be more robust of one or two other DNA extraction methods were performed side-by-side. Do other DNA extraction kits that perform well with EVs isolated from conditioned culture media (e.g. QiaAMP DNA Micro; XCF Exosomal DNA) also perform as well as SeleCTEV? This would be a more comprehensive analysis of the performance of extracted EV-DNA in mutation analysis from patient plasma samples.

Minor concerns:

11.  In their mass spectrometry analysis of EV markers from the EVs of CFPAC cells (Figure 1D) the authors fail to identify CD63 or CD81 in these EVs; CD63 and CD81 are major tetraspanin components of EVs and the authors are also able to identify CD81 by Western blot analysis from CFPAC EVs (Figure 1C), a less sensitive detection method. The authors should comment on this discrepancy.

Author Response

Reviewer 2:

We thank the reviewer for the constrictive suggestions. Point-by-point responses are provided bellow. Please also find attached the revised manuscript.

In their manuscript titled “Defining optimal conditions to tumor Extracellular Vesicle DNA extraction for mutation profiling” Elzanowska et al. performed a comparative analysis of five different DNA extraction methods/kits for the extraction of EV-DNA from EVs that have been isolated by ultracentrifugation from conditioned cell culture media. The authors show that each method/kit has different advantages or disadvantages when it comes to DNA yield, DNA size, and suitability in downstream assays such as TaqMan qPCR and next-generation sequencing. The authors then use one method for DNA extraction from plasma EVs for use in mutation detection. This is a useful study for the EV field to help investigators choose the most appropriate DNA extraction methods for their studies; however, there are some outstanding questions that should be addressed.

Major concerns:

  1.  One of the DNA extraction kits chosen by the authors, the SeleCTEV Exosomal DNA Kit, is designed not only to be used for DNA extraction from EVs but also for EV isolation from a biofluid. For the SeleCTEV kit the EV isolation is performed using an included proprietary peptide. It is unclear from the Methods section whether the authors included the peptide step in their DNA extractions or if they simply omitted the peptide-mediated EV isolation steps for their workflow.

Reply: We omitted the peptide-mediated EV isolation step of SeleCTEV Exosomal DNA Kit protocol in order to normalize EV samples for DNA extraction and used only the part of the protocol for DNA purification, which is now mentioned in 2.5 DNA extraction, Page 5, Lines 233-235. We also highlight the use of only one method for isolation of EV samples in Discussion, Page 16, Lines 625-630 which reads as: as a first layer of standardization, we fixed the EV isolation method used, so that all DNA would be extracted from comparable isolated EV samples. The chosen method for the isolation of EVs was differential ultracentrifugation coupled with sucrose cushion, which is one of the most commonly used EV isolation techniques and is considered a gold standard for EV isolation. Other EV isolation methods were not compared, as it is beyond the scope of this paper.

  1. Moreover, it would be interesting (and important) to determine if the SeleCTEV Exosomal DNA Kit performs similarly for DNA isolation when the proprietary peptide that is included is used for EV isolation, rather than ultracentrifugation. This would be important information for the EV community as this kit could potentially be used for all steps – EV isolation and DNA extraction – in study workflows.

Reply:  In our work, for standardization purposes we fixed the EV isolation method used so that all DNA would be extracted from comparable isolated EV samples. However, we added information in Discussion, Page 18, Lines 728-733 that it was previously suggested that SeleCTEV’s proprietary peptide for plasma EVs isolation improves EV isolation for DNA studies when compared to ultracentrifugation (Zocco, D., et al., Isolation of extracellular vesicles improves the detection of mutant DNA from plasma of metastatic melanoma patients. Sci Rep, 2020. 10(1): p. 15745.10.1038/s41598-020-72834-6). Therefore, future research should consider the potential benefits of using SeleCTEV’s proprietary peptide for improved plasma EV-DNA study. Due to the limitation of plasma sample availability, side-by-side comparison of other DNA extraction methods was not possible in our study.

  1. For their plasma samples and DNA mutation analysis the authors chose to only use the SeleCTEV kit for DNA extraction (Table 5). While their data show that DNA extraction with this kit permits identification of all mutations present in the tumour tissue, this analysis would be more robust of one or two other DNA extraction methods were performed side-by-side. Do other DNA extraction kits that perform well with EVs isolated from conditioned culture media (e.g. QiaAMP DNA Micro; XCF Exosomal DNA) also perform as well as SeleCTEV? This would be a more comprehensive analysis of the performance of extracted EV-DNA in mutation analysis from patient plasma samples.

Reply: Due to the limited availability of plasma samples, as well as to ensure sufficient amounts of material for mutational analyses, we only had enough material to isolate EV-DNA using the SeleCTEV Exosomal DNA Kit, which gave the highest DNA yield when analyzing CCM-derived EVs (Discussion, Page 18, Lines 726-728; 732-733).

Minor concerns:

  1. In their mass spectrometry analysis of EV markers from the EVs of CFPAC cells (Figure 1D) the authors fail to identify CD63 or CD81 in these EVs; CD63 and CD81 are major tetraspanin components of EVs and the authors are also able to identify CD81 by Western blot analysis from CFPAC EVs (Figure 1C), a less sensitive detection method. The authors should comment on this discrepancy.

Reply: We added a comment regarding this result in 3.1 Characterization of EVs isolated from conditioned media, Page 8, Lines 358-371, where we explain the potential reasons for not detecting CD63 and CD81 in some of the EV samples analyzed by LC-MS. Firstly, both tetraspanins are membrane proteins, which are both hydrophobic (difficult to solubilize) and low-abundant, which makes them difficult to detect with our LC-MS method. Moreover, the scan speed of any mass spectrometer is not sufficient to detect all peptides in a sample, so some peptides will not be detected even if they are present in sufficient amount. If the peptides from these proteins elute from the LC and enter the MS at the same time with many other peptides that are present in higher concentrations, these higher-abundant peptides will be prioritized in MS identification, and the lower-abundant ones will be missed. Interestingly, a comprehensive analysis of EV protein markers across more than 100 different human tumor and non-tumor cell lines showed that only 40% of them are positive for CD63 (Hoshino, Ayuko et al. “Extracellular Vesicle and Particle Biomarkers Define Multiple Human Cancers.” Cell vol. 182,4 (2020): 1044-1061.e18. doi:10.1016/j.cell.2020.07.009).

Round 2

Reviewer 1 Report

Authors have definitely improved the manuscript. Overall, the study contributes to build the know how on EV-based liquid biopsy.  

Reviewer 2 Report

The authors' changes to the text of the manuscript have adequately addressed my concerns and their explanations for not performing the suggested additional experiments are reasonable.